# Influence of Carbonated Bottom Slag Granules in 3D Concrete Printing

**DOI:** 10.3390/ma16114045

**Published:** 2023-05-29

**Authors:** Karolina Butkute, Vitoldas Vaitkevicius, Maris Sinka, Algirdas Augonis, Aleksandrs Korjakins

**Affiliations:** 1Faculty of Civil Engineering and Architecture, Kaunas University of Technology, Studentų g. 48, 51367 Kaunas, Lithuania; vitoldas.vaitkevicius@ktu.lt (V.V.); algirdas.augonis@ktu.lt (A.A.); 2Institute of Materials and Structures, Riga Technical University, Kipsala Street 6A, LV1048, Riga, Latvia; maris.sinka@rtu.lv (M.S.); aleksandrs.korjakins@rtu.lv (A.K.)

**Keywords:** bottom slag, burnt fly ash, 3D printing, CO_2_ reduction, artificial aggregate

## Abstract

This study investigates the possibility of utilising bottom slag (BS) waste from landfills, and a carbonation process advantageous for the use of artificial aggregates (AAs) in printed three-dimensional (3D) concrete composites. In general, the main idea of granulated aggregates is to reduce the amount of CO_2_ emissions of printed 3D concrete objects (wall). AAs are made from construction materials, both granulated and carbonated. Granules are made from a combination of binder (ordinary Portland cement (OPC), hydrated lime, burnt shale ash (BSA)) and waste material (BS). BS is a waste material left over after the municipal waste burning process in cogeneration power plants. Whole printed 3D concrete composite manufacturing consists of: granulating artificial aggregate, aggregate hardening and sieving (adaptive granulometer), carbonation of AA, mixing 3D concrete, and 3D printing. The granulating and printing processes were analysed for hardening processes, strength results, workability parameters, and physical and mechanical properties. Printings with no granules (reference 3D printed concrete) were compared to 3D printed concretes with 25% and 50% of their natural aggregate replaced with carbonated AA. The results showed that, theoretically, the carbonation process could help to react approximately 126 kg/m^3^ CO_2_ from 1 m^3^ of granules.

## 1. Introduction

The conversion of waste (municipal solid waste (MSW)) to energy in cogeneration power plants has grown over recent decades. About 500 cogeneration power plants are already operating around the world [1,2,3,4,5]. Burning waste is a better solution, because 1 ton of waste produces 1 ton of CO_2_, contrary to landfill waste that produces methane after anaerobic decomposition of the biodegradable portion of the waste, which, as an equivalent, produces 1.38 tons of CO_2_ [6,7]. Furthermore, after the cogeneration process of burning MSW, residual material is left behind—BS. BS is the residue originating in the grate furnace of a municipal solid waste incinerator (MSWI). Remaining incombustible, residues represent 5–10% by volume of MSW streams [8].

Above all, the carbonation method (in a climate chamber), or CO_2_ curing, is founded on the basis of obtaining denser carbonate, which not only improves the physical and mechanical properties of cement-based materials, but also has the ability to utilize and store CO_2_ safely in AAs, and permanently in whole 3D printed objects, without the carbonation of large construction projects [9,10]. Accelerated carbonation is applied directly to the matrix, which could positively affect carbonation of the granules. CO_2_ capturing technology already exists, and has potential for lowering energy consumption, which could have an advantage in a sustainable circular economy [9,11,12,13].

Many researchers are searching for a way to reduce OPC usage and minimize CO_2_ emissions during OPC production. For example, 1 kg of OPC produces 0.9–1.0 kg CO_2_. Carbonation is one of the ways to reduce CO_2_ emissions without minimizing the amount of OPC used in 3D printed concrete. This technology is simpler when only small materials, such as AAs, are carbonized, instead of the entire printed object. In the building industry, experiments were conducted in the printing of small objects (bench, footbridge, and decorative walls) and small-scale buildings (one floor or shelters), which have the possibility to be carbonated after printing. However, for objects with massive dimensions (two-story buildings, or taller than 9 or 200 m), there is little possibility for additional carbonation after printing [14,15,16,17].

When investigating the properties of 3D printed concrete, the rheology, extrudability, flowability, and buildability become the main factors [18]. All of these mentioned parameters were tested with the various materials used in 3D printing composites (superplasticizer, viscosity modifying agents, defoaming agents, accelerators) [19].

## 2. Materials and Methods

### 2.1. Binder Materials

This study investigated two different objects: handmade AAs (granules), and 3D printed composites (3DPCs). Both of them were made with the same binder materials. The main binder material was OPC CEM I 42.5 R from the local manufacturer AB “Akmenės cementas” (Lithuania, Naujoji Akmenė). Another binder material was hydrated lime (HL) (calcium hydroxide), class CL 90-S from “Lhoist Bukowa Sp. z o. o.” (Poland, Bukowa). As a secondary product from oil shale power plants, burnt oil shale ash (CON BS) from “Enefit Power AS” (Estonia, Auvere) was used in granules and 3D printed composites. The burnt ash chemical composition shows that it also has CaO, which could be carbonated in natural conditions and in a CO_2_ chamber.

From an ecological aspect, AA production used 0–5 mm particle diameter BS from UAB “Kauno kogeneracinė jėgainė” (Kaunas cogeneration power plant) (Lithuania). Slag consists of mostly unreactive material (only 20.14% of CaO), and is more like an inert material. For the purpose of obtaining initial strength before carbonation, binders were added to all of the granule composites, as mentioned previously (OPC, HL, CON BS). All of the binder chemical compositions and densities are mentioned in Table 1. In general, bottom slag is mainly composed of noncombustible materials such as stone, glass, ceramic, sand, and metal. Lateral elements were refined after the incineration process, before storage in landfills, by a special magnetic sieving system. Figure 1 shows an SEM image of BS at one hundred times magnification. 

In Figure 2, the chemical elements in BS are analysed. Including the fact that MSW is highly varied, BS chemical elements could also be diverse. This research only shows the possibility of using previously analysed BS samples, and there is no guarantee that other BS samples will have the same composition of chemical elements. In a situation like this, we engage the hypothesis that the carbonation of calcium oxide and calcium carbonate could lock the BS elements, and homogenize the granule properties in 3D printed concrete.

### 2.2. Aggregate and Additives

Without the previously mentioned binder materials, 3DPC consists of other aggregates and additives. The main natural aggregate used in the composites was sand from the UAB “Rizgonys” (Lithuania, Rizgonys) quarry, which was dried and sieved into particle sizes ranging from 0 to 2 mm. To control the setting and hardening time of the 3DPC, an accelerator—calcium formate from “Mudanjian Fenga Chemicals Imp. & Exp. Corp” (China, Mudanjiang), was added. Another additive to regulate expansion was Denka, from “Neuvendis SPA” (Italy, Milan). As a dry plasticizer in composites, Peramin was used, which is a sulfonated melamine polymer, from “Imerys S.A.” (France, Fos-sur-Mer). To achieve a hydrophobic effect and adhesion between layers, the hydrophobic dispersive polymer powder Vinnapas, from “Wacker Chemie AG” (Germany, Burghausen), was used. For reinforcement and reduction of cracks, polypropylene fiber (3 mm length) “Belgian fibers manufacturing” (Belgium, Kortrijk) was added.

### 2.3. Test Methods

#### 2.3.1. Granulation

Each year, landfills around the world accumulate millions of tons of BS. One solutions to eliminate BS could be its usage as a supplementary cementitious material in concrete mixtures [20]. BS from landfills could be used, when crushed, ground, or sieved (e.g., 0–2 mm; < 2.36 mm) [21,22]. Likewise, scientists from China suggested using BS as a replacement for recycled fine aggregate in mortar, which gives positive but debatable results [23]. Similarly, some natural aggregates in green concrete could be replaced by MSWI BS manufactured lightweight coarse aggregates. In another specially treated method, with dry and wet treatment processes, granulated BS could replace up to 100% of the natural gravel in concrete, be used for road base layers, or be used as a secondary aggregate in asphalt applications [24]. Lightweight aggregate made from BS could be made in single-step or double-step pelletization processes, in diameters of 4.75–9.5 mm, 9.5–16 mm, 5–10 mm, 6–12 mm, 8–16 mm, 2–8 mm, 0–4 mm, 0–19 mm, and 2–16 mm [21,22,24,25,26,27,28]. In this system, AA was made with a mechanical process using the agitation granulation method. This cold bonding process occurs when moisturized materials are rotating in a disk granulator, without an external compacting force (Figure 3). The quality of these granules depends on the amount of fine particles in the dry materials portion. A sufficient amount of fine material particles could ensure good granule formation (Figure 4). 

The disk granulator has the following parameters: disk diameter (D)—500 mm, disk height (H)—100 mm, inclination angle (α)—45°, revolution speed (n)—35 rounds/min, load size—1.4 kg.

For this research, three different designs of artificial aggregate were developed. After granulation, fresh granules were spread in a thin layer on a horizontal surface for natural drying, at 20 ± 2 °C and 50 ± 5% relative humidity, for 2 days. After drying, granules were fractionated through a 4 mm wire mesh sieve to a 0–4 mm fraction. Larger fraction granules (< 4 mm, about 20% of the whole volume) were not used in other tests. Sieved granules were tested for particle size determination. Specifically, the granule diameter was chosen after comparison with the natural aggregate diameter in reference 3D printed concrete, and also after consideration of the 3D printing laboratory equipment and the purpose of the printed object.

#### 2.3.2. Carbonation

The carbonation process occurs in materials or composites which have sufficient amounts of calcium hydroxide or calcium oxide, which is required for the reaction of these two compounds with CO_2_ [30]:Ca(OH)_2_+CO_2_→CaCO_3_+H_2_O(1)
CaO+CO_2_→CaCO_3_(2)

More importantly, after carbon dioxide treatment in a climate chamber, granules have more calcium carbonate, as a result of this fact, granules have increased strength. The carbonation process could also affect the water absorption parameters of granules.

In this research, a complex climate chamber carbonation technology, without pressure, was used. The scheme is shown in Figure 5. Throughout the entire carbonation process, the climate chamber was set at 20 ± 2 °C, 70 ± 5% relative humidity, and 20 ± 3% CO_2_ concentration. As other scientists claim, curing temperatures ranging between 20 and 80 °C yield little difference in compressive strength [31]. Other external investigations have proven that a relative humidity of 50–65% is optimal for carbonation reactions, because a higher or lower percentage of humidity could slow down the carbonation reactions. That is why most of the carbonations, including this research, have curing conditions of 20% CO_2_ concentration with 70% relative humidity. This is the average range of humidity and temperature where the carbonation of aggregate (or another concrete element) reaches the innermost depth [20,32].

The titration method was used for the additional calculation of reacted lime. One g of ground granules, AA2, was poured into 150 mL of distilled water and dissolved, and 3 drops of 1% phenolphthalein indicator solution were added. A pink color showed that the solution contained soluble alkali materials at a pH greater than 10. Later, HCl (acid of salt) was titrated until calcium hydroxide reacted, and the solution became transparent. The amount of calcium oxide in the solution was calculated based on the amount of added acid of salt:𝐴 = ((𝑉⋅𝑇_𝐶𝑎𝑂_)/*m*)⋅100(3)

*V*—1N HCl amount used for titration;

𝑇_𝐶𝑎𝑂_—1N HCl acid titre, expressed CaO amount, (2.804 g/mol);

*m*—ground granules mass, g.

Uncarbonated granules have 177 kg of CaO per 1 m^3^ of bulk granules. After 1 day of carbonation in the climate chamber, the amount of CaO in the granules was reduced to 118.7 kg per 1 m^3^ of bulk granules. After 3 days of carbonation, the amount of CaO was reduced to 6.2 kg per 1 m^3^ of bulk granules. These results show that the longer the carbonation process extends, the more CaO reacts and forms CaCO_3_.

#### 2.3.3. Tests for Artificial Aggregate

After the granulation process, granules were tested, according to the adapted lightweight aggregate bulk crushing resistance test, to determine aggregate crushing strength. The test was performed partially according to the EN 13055-1 standard [34], to compare the obtained parameters with those of other researchers [11]. Granules were poured into a cylinder and vibrated for 5–10 s. The cylinder was filled to the upper line, if needed, after vibration. Then, the piston was put on top of the granules. The entire testing apparatus was built into a hydraulic press (Figure 6). The aggregate strength was fixed when the piston, on top of the granules, moved down to the marked boundary [34,35,36].

Then, the artificial aggregate particle density was measured, to define the artificial aggregate type. The 10 most rounded artificial aggregate granules were selected, and dried until constant weight in a laboratory heating oven at 105 ± 1 °C. After drying, each granule was covered with thin layer of wax and weighed with hydrostatic scales. Formula (4) was used for the calculation of particle density [37].
(4)$$\rho_{d}=\frac{m}{m_{p}-m_{pv}-\frac{m_{p}-m}{\rho_{p}}}$$

$\rho_{p}$—density of paraffin, 930 kg/m^3^;

*m*—granule mass, kg;

$m_{pv}$—granule mass covered with paraffin in water, kg;

$m_{p}$— granule mass covered with paraffin, kg.

A scanning electron microscope (SEM), Hitachi S-3400N, and energy dispersive spectrum (EDS) methods were used for the microscopic structure analysis of artificial aggregate and printed 3D elements.

The moisture content in granules was measured according to the EN 1097-5 standard [38]. First, a sample of granulated AA was weighed (*m*_1_). Then, the sample was put into a thermal oven (Memmert) and dried for 24 h at 65 °C. After drying, the sample of granules was taken out and immediately weighed (*m*). The moisture content of the ready-made granules was calculated using Formula (5) [38].
(5)$$W_{m}=(\frac{m_{1}-m}{m})\cdot 100$$

To measure water absorption, artificial aggregate was immersed into a water container until the mass was constant. The test portion was removed from the water, immediately dried using an absorbent cloth, and immediately weighed (*M*_1_). Then, the test portion was oven dried at 110 ± 5 °C until it reached constant mass (*M*_3_) [39]. The calculation was performed using Formula (6).
(6)$$W_{cm}=\frac{M_{1}-M_{3}}{M_{3}}\cdot 100$$

#### 2.3.4. Tests for Fresh State Composition

The consistency of freshly mixed composite was measured according to the flow table test in the EN 1015-3 standard (Figure 7a) [40]. After this test, the bulk density was measured in a 1 L bowl according to the EN 1015-6 standard (Figure 7b) [41]. Then, a device for measuring air content was attached to the same bowl according to the EN 1015-7 standard (Figure 7c) [42]. Lastly, after testing the properties of the fresh mix, prisms (4 × 4 × 16 cm) were formed for strength measurement.

#### 2.3.5. Three-Dimensional Printer

Three-dimensional printing was carried out using a custom-made gantry-type printer, developed within Riga Technical University (RTU), for printing building materials, such as concrete. The printer has an aluminium frame with dimensions of 2000 × 1000 × 1200 (h) mm, allowing for maximum model dimensions of 1500 × 1000 × 1000 mm, and a hopper volume of up to 30 L. Control of the printer was achieved using open-source Repetier-Firmware, while slicing was performed using Simplify3D 5.0 software by Simplify 3D Ltd. in the U.S.A. For creating the 3D models, Solidworks 3D CAD design software was used. The printer head nozzle has a diameter of 20 mm, a layer width ranging from 35 to 45 mm, and a layer height of 10 mm. The printer construction is shown in Figure 8 [43].

#### 2.3.6. Three-Dimensional Printing and Curing

In the printing laboratory, the temperature during the entire mixing, printing, and curing process was maintained at 18–20 °C, with a relative air humidity of 30 ± 5% as measured by a hydrometer. A portable mixer, Rubimix-9 by Rubi UK LTD, was used for mixing at an average speed of 780 RPM. First, all dry composite materials were weighed and mixed before the printing process. Next, the prepared dry composites were poured into a mixing tub and mixed again with mixer for 10–15 s to ensure homogeneity. Then, tap water (at a temperature of 9–10 °C) was poured into the composite mixture and stirred for 100–120 s. A portion of the mixture was taken for testing of the fresh mortar properties and prism formation in the moulds, while the remaining composite was poured into the printer’s hopper and extruded through the printer head and nozzle until a homogenous mass of concrete started to flow. Elements of 160 × 160 × 200 (h) mm and 160 × 160 × 120 (h) mm were printed at a speed of 100 mm/s. All printed objects were kept under the same conditions as during the printing process for 28 days (Figure 9 and Figure 10). Afterward, the printed elements were cut into pieces using a concrete angle grinder for testing of the mechanical properties.

Different water contents in the composites affected their printability. In an experimental approach, water was added to the 3D printing mixes to obtain a mixture that could be printed under the same printer conditions and parameters. It was observed that increased water content in the composites was due to the larger mass granules that replaced the sand, and the different water absorption properties of the granules.

#### 2.3.7. Tests for Printed Objects

After mixing the 3D printing composites with water, part of the mixture was formed into prisms (moulded) according to the EN 1015-11 standard [44]. Following the mentioned standards, the prisms were cured under laboratory conditions at 20 ± 2 °C and 30 ± 5% relative air humidity for 28 days. Another part of the mixture was poured into the printer’s hopper and used to print objects. After 28 days of curing under the same laboratory conditions (20 ± 2 °C, 30 ± 5% relative air humidity), the printed elements were cut with a concrete angle grinder and, along with the prisms, were measured for flexural and compressive strength with a compression and bending testing machine (“ratio TEC”). Descriptions of the measurement method are in Figure 11 [45].

## 3. Results

### 3.1. Artificial Aggregate Test Results

After the analysis of AA test properties, and other research, granule binder materials were selected, as shown in Table 2. These compositions resulted in handmade AA—granules for 3D printing. Figure 12a shows ready-made granule particles.

After first sieving through a 4 mm sieve (Figure 12b), all AA configurations were sieved again, to compare the particle size distribution between aggregates. Approximately 10–15% of the artificial aggregate was not used in 3D printing after sieving, because the particles were larger than 4 mm in diameter. That unused portion could be used in other elements of concrete (printed object). Table 3 shows the percentage of sieved residual portions after particle size determination. From the results, it can be seen that the percentage of portions from 2.8 mm to 1.0 mm sieves are very similar when comparing all three samples.

AA was poured and weighed in a 1 L bowl, only by gravitation. Table 4 shows the measured AA bulk densities before and after the carbonation process. From the results, it can be seen that not all AA compositions obtained a positive density after carbonation. Still, AA2 and AA3 show promising results.

The moisture content of the granules was measured according to the EN 1097-5 standard [38], and the results are shown in Figure 13.

As Figure 13 shows, the carbonated granules mostly have reduced water content, as is expected. This also shows that the granules in AA1 and AA3 have more open pores for water evaporation in the drying process.

The artificial aggregate water absorption measurement results are displayed in Figure 14.

The test results show that different amounts of granules demand different amounts of water (Figure 14). From both Figure 13 and Figure 14, it can be noticed that AA2 granules have a larger water immersion potential.

Table 5 shows the sieved (0–4 mm) AA density difference between carbonated and uncarbonated particles. From the density measurement results, it can be seen that the carbonated aggregates have diversity, because not all aggregate compositions display the same correlation of results. The AA1 composition reveals that, after carbonation, the combination of BS and Portland cement does not form denser elements. On the contrary, the AA2 and AA3 compositions showed apparently different results. Compared with the initial density, the density after carbonation increases about 14% in the AA2 composition, and about 10% in the AA3 composition.

Comparing the AA crushing strength results in Table 6, it can be seen that granules with a calcium hydroxide binder have a smaller increase in strength than granules with burnt fly ash and Portland cement as a binder. This could be because of different amounts of reactive magnesium oxide in the binders, which, after hydration, absorb CO_2_ and form stable hydrated magnesium carbonates that densify the microstructure and gain strength [27,30].

The crushing strength was increased, and in one instance was even more than 150%. This could be because of calcium oxide minerals reacting with CO_2_, which form calcium carbonate crystals to improve the strength of the granules. Including this fact, carbonated aggregate could have an adverse effect on the carbonization resistance of 3D printed concrete elements. There could be a higher possibility of the corrosion of steel bars. In 3D printed concrete construction, steel reinforcements are infrequently used, which is why the carbonization level could be excluded from evaluation. Artificial aggregate could absorb even more CO_2_, and is a great possibility for reducing carbon emissions [32].

After analysis of all the artificial aggregate test results, it was decided to print 3D concrete elements only with the granules showing the most promising results, AA2 and AA3. Further analysis will utilize the previously mentioned carbonated granules. For the chosen AAs, microscopic analysis was performed with an SEM to identify what could be expected to happen after printing, e.g., adhesion of the cement paste with the AA. Figure 15 shows SEM images of the granules selected for 3D printing; carbonated and uncarbonated AA granules were split open to analyze the aggregate structure, porosity, and uniformity. All granules EDS analysis were given in Table 7, for comparison of elements inside and only on surface. 

When comparing carbonated and uncarbonated granules (AA2), EDS analysis showed that the carbon and oxygen content increased after the carbonation process.

A high-resolution powder X-ray diffractometer (XRD) was used to measure the carbonate content in the AAs. The results in Figure 16 and Figure 17 show that the main components in the AAs are quartz and calcite, which means that in the AAs, a reaction with CO_2_ happens after carbonation.

### 3.2. Three-Dimensional Printing Compositions

Etalon composition was mixed with 25% carbonated and uncarbonated granules, after that formed prisms and tested strengths. Table 8 shows the results, which demonstrate the advantage of carbonation, not only for granules, but also for increasing the composite’s strength when granules are in it. These results reflect the fact that when CO_2_ reacts with calcium hydroxide, calcium carbonate is formed, which increases hardness and decreases porosity [46].

Table 9 shows information about the 3D printed concrete compositions. All materials were weighed by mass. The additives in 3D concrete composites are: accelerator, superplasticizer, polypropylene fiber, shrinkage reducing additive, and dispersible powder. Compositions LV2 and LV3 have 25% of their natural aggregate replaced with AA. Composition LV4 has 50% AA replacement.

### 3.3. Results after 3D Printing

In Table 9 mentioned 3D printing composites were mixed with water, and measured fresh state parameters (Table 10). For the same composites after 28 days of hardening were measured flexural and compressive strength parameters (Table 11).

Figure 18 shows cross-sections of the printed and cut concrete elements, how perfectly the layers are connected to each other, and how the AA is distributed throughout the layers and the entire structure.

Prisms (4 × 4 × 16 cm) were formed from all four 3D printing composites. The prisms were cured for 7 days at 20 ± 2 °C and 90 ± 5% relative humidity. Immediately after splitting the prisms open, the carbonation test was performed according to the LST EN 14630 standard. All prisms were sprayed with a thin layer of 1% phenolphthalein reagent (diluted in 30% ethanol and 70% distilled water). Carbonated areas can be seen in Figure 19.

Figure 19 shows the carbonation test results of 3D printed concrete composites with carbonated and uncarbonated granules. Clear areas show carbonated places, and colored areas show uncarbonated places. Inside the prisms, the AA appears purple only in the places where the aggregate was not carbonated, or alkaline. Areas with carbonated granules show no color, which means it is carbonated.

SEM images of the printed object crosscuts have been analysed to investigate the microstructure of the printed elements, and especially the connection zone between the cement paste and aggregates. Figure 20b shows the natural aggregate (NA) connection zone with the cement matrix. It can be seen that the NA surface is smoother than the granule, and almost no cement matrix is attached to the surface. The connection zone in Figure 20a shows a crack.

Figure 21b shows the interface between carbonated AA and the cement matrix. In this image, the connection zone looks bonded. From the split printed object, c and d were found in places where AA was bonded, but hard to notice. As noted in the images, the interface between the granule and cement matrix is more solid than with the natural aggregate. The reduced strength of 3D printed composites could be because natural aggregate has greater particle strength. The rough surface of the AA is seen in image f, which could be because of residue of the cement hydration products. This increases the bonding strength between the artificial aggregate and the cement matrix. From image e, it can be seen that the connection zone between the AA and cement paste is very small, and probably exists only because of the splitting force used after the flexural strength measurement.

## 4. Conclusions

The granulator-made artificial aggregate, after measurement, was determined as a fine-grained, porous, lightweight aggregate (natural sand particle density approximately 2650 kg/m^3^, AA—1610–1670 kg/m^3^).The tested carbonated granules showed the advantage of the carbonation process. When comparing carbonated and uncarbonated granules, in reference to 3D printed concrete compositions, it was noticed that compression strength was increased from 3 to 9%.Granules with a calcium hydroxide binder showed inferior strength results compared to granules with OPC and BSA as a binder. Lateral granules have better water absorption properties and lower moisture content, but granules with a calcium hydroxide binder have greater particle density and aggregate crushing strength compared with AA made only with OPC as a binder.AA granules made with MSW BS were first carbonated in a climate chamber (without pressure), and then added to 3D printing concrete composite for printing. This method shows that the whole printed object could have lower CO_2_ emissions because of AA usage, and there is no need to carbonize the entire object.Theoretically calculating, carbonated granule (AA2) usage could reduce (sequester) about 126 kg of CO_2_ from 1 m^3^ of granules, and form CaCO_3_.Three-dimensional printing composites containing carbonated granules, after replacing 25% of mass volume natural aggregate with AA, reduces compressive strength results by 1–13%. After 50% natural aggregate replacement with AA, compressive strength results were reduced by almost 29%. Despite this fact, the strength results still maintain a safe margin for construction element stability, and still correspond to the technical requirements for private house buildings.

## Figures and Tables

**Figure 1 materials-16-04045-f001:**
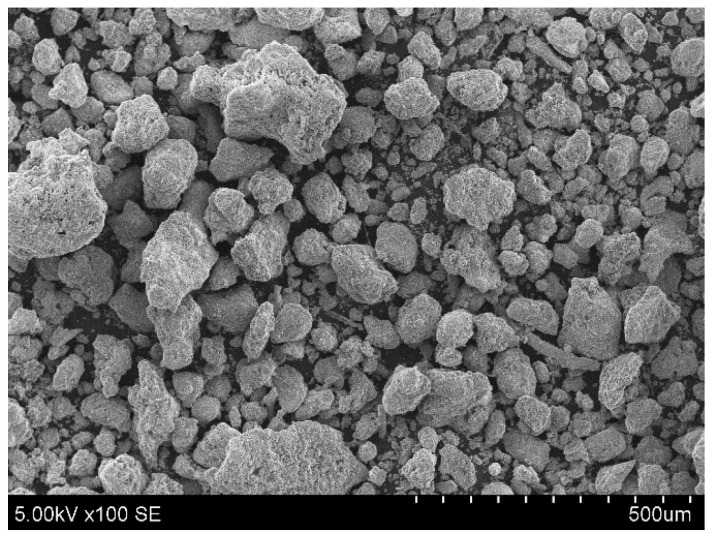
BS SEM image (×100).

**Figure 2 materials-16-04045-f002:**
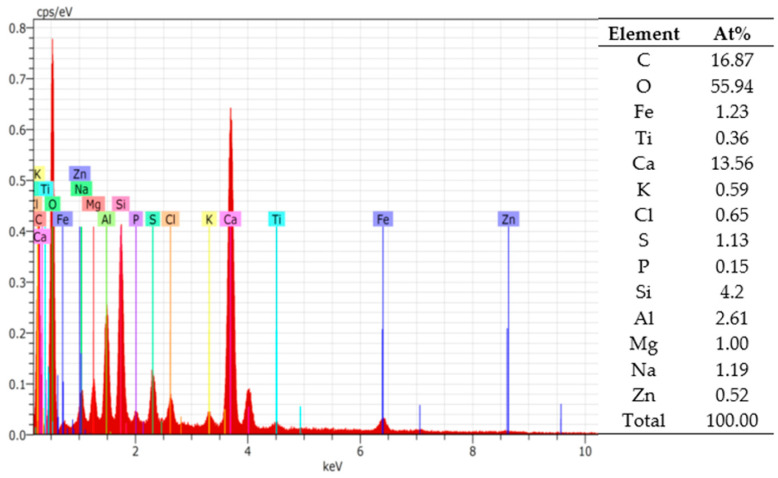
EDS spectrum analysis of BS.

**Figure 3 materials-16-04045-f003:**
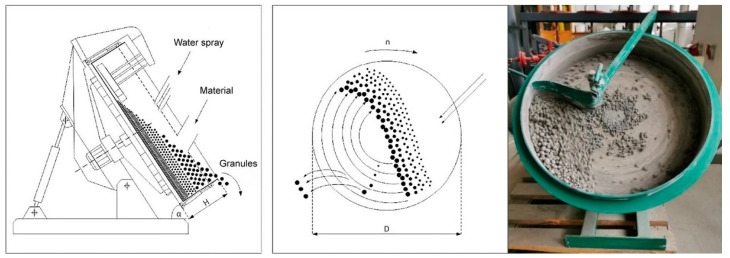
Disk granulator scheme and photo [29].

**Figure 4 materials-16-04045-f004:**
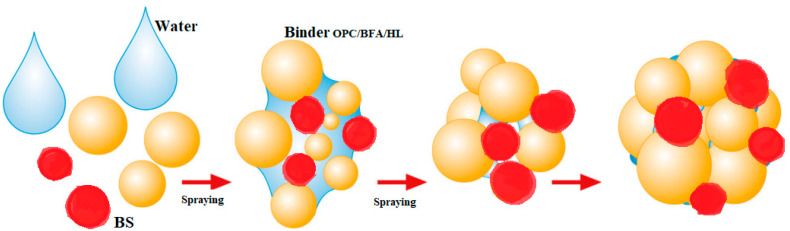
Schematic diagram of AA formation in the granulation process [25].

**Figure 5 materials-16-04045-f005:**
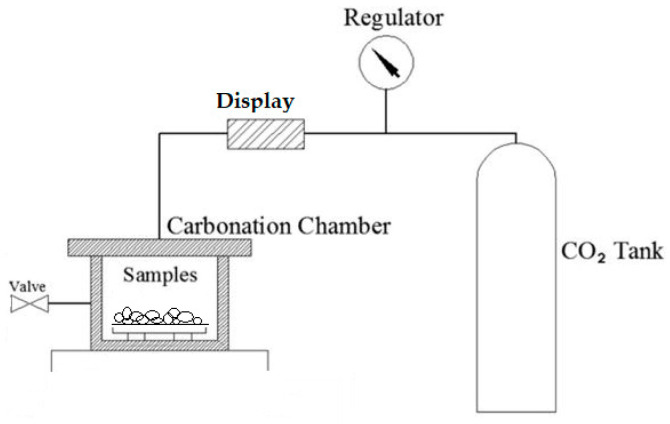
Carbonization chamber scheme [33].

**Figure 6 materials-16-04045-f006:**
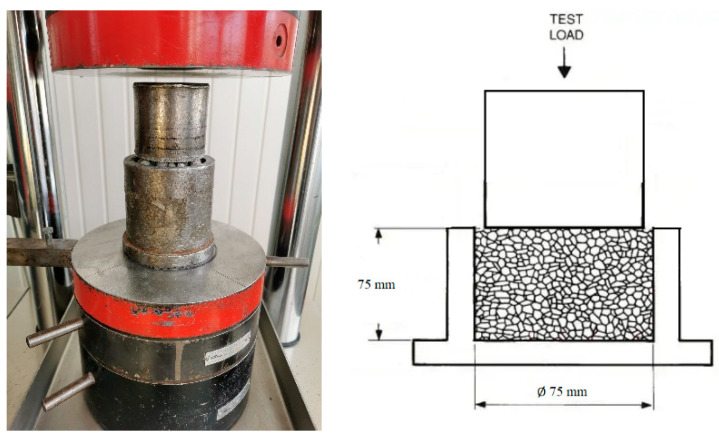
Aggregate crushing strength measurement device and scheme.

**Figure 7 materials-16-04045-f007:**
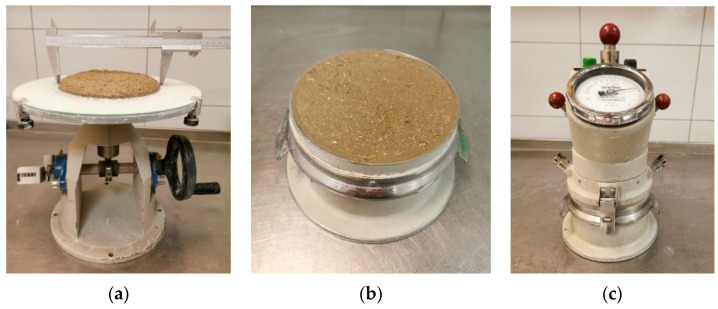
Fresh 3D printing composite test equipment: flow table (**a**), 1 L bowl (**b**), air content measuring device (**c**).

**Figure 8 materials-16-04045-f008:**
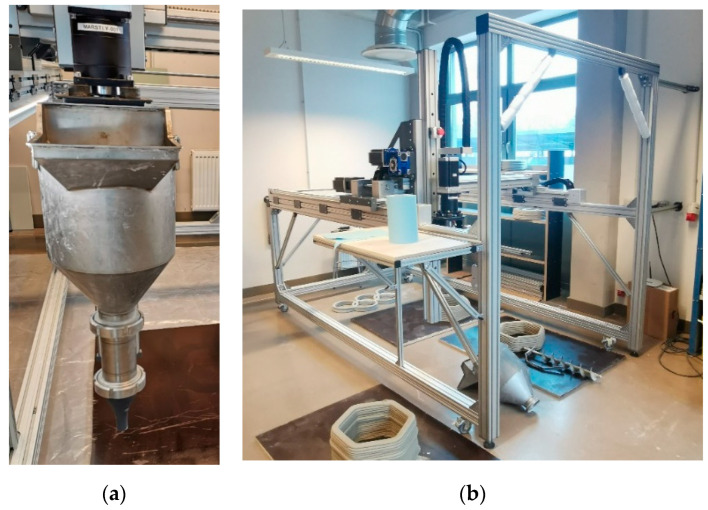
Three-dimensional printer batch-type print head with hopper (**a**), and 3D printer (**b**).

**Figure 9 materials-16-04045-f009:**
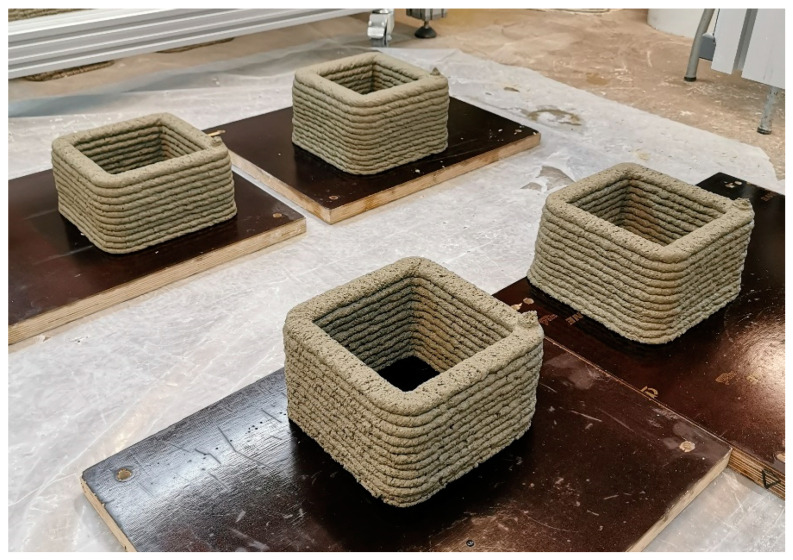
Printed objects.

**Figure 10 materials-16-04045-f010:**
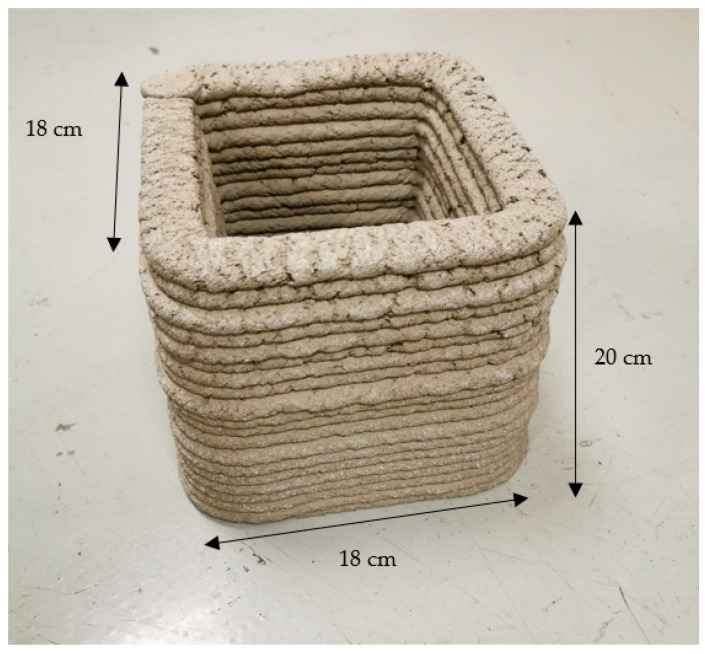
Printed object.

**Figure 11 materials-16-04045-f011:**
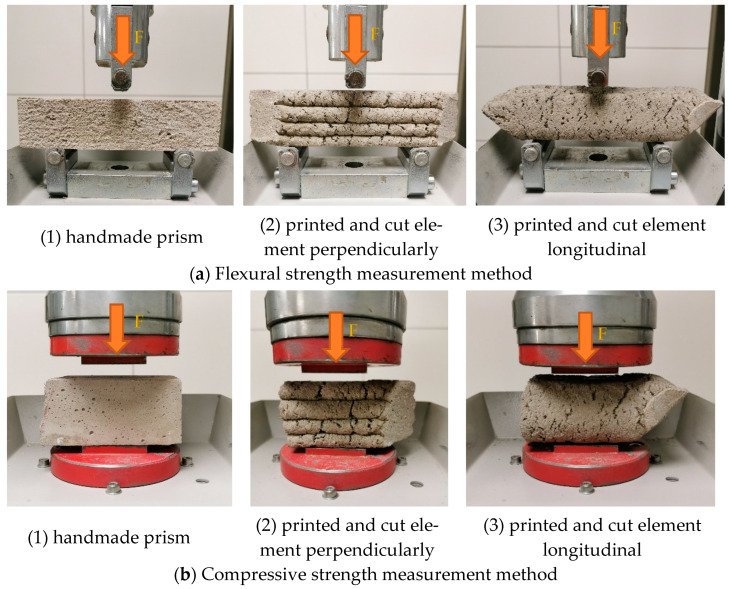
Flexural and compressive strength measurement methods of 3D printed objects (F-force of compression).

**Figure 12 materials-16-04045-f012:**
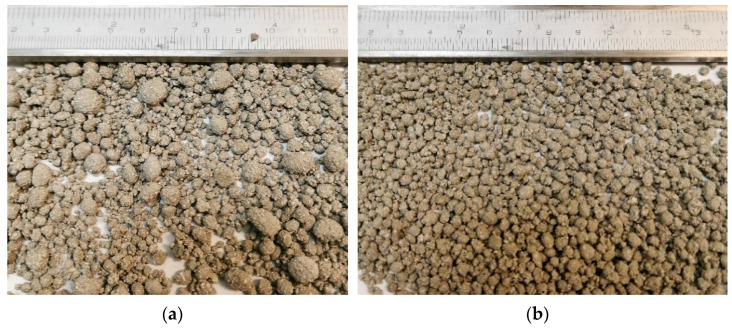
AA granules before sieving (**a**) and after sieving with a 4 mm sieve (**b**).

**Figure 13 materials-16-04045-f013:**
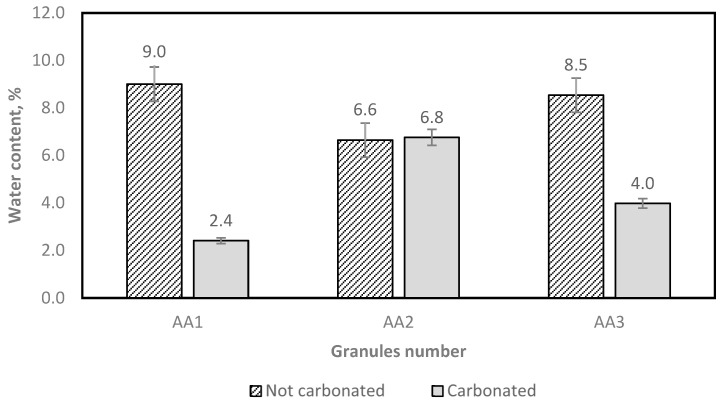
Determination of the aggregate water content after oven drying.

**Figure 14 materials-16-04045-f014:**
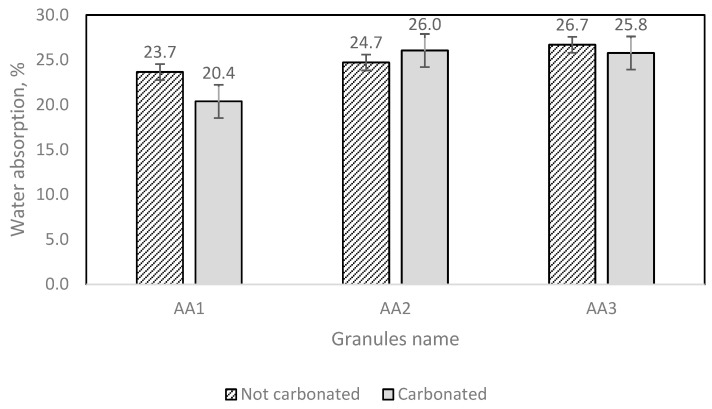
AA water absorption results.

**Figure 15 materials-16-04045-f015:**
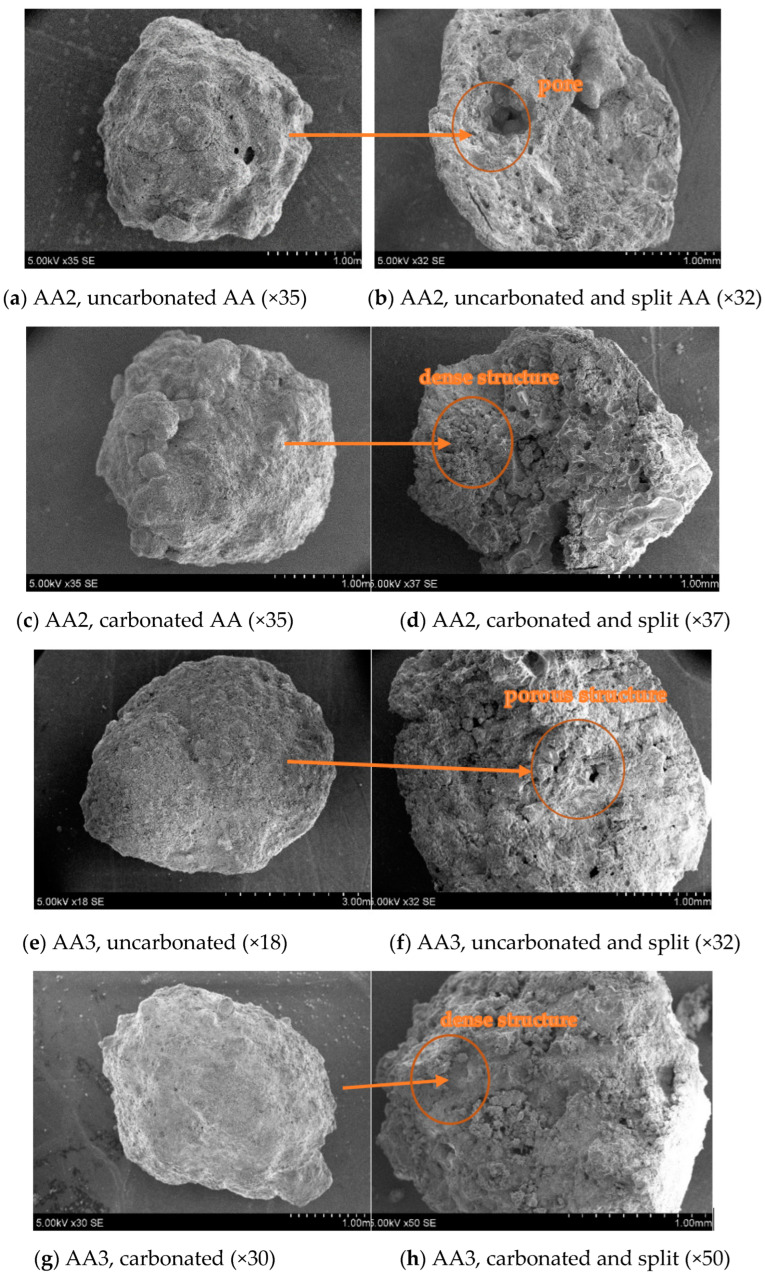
Microstructure of AAs visualized with SEM.

**Figure 16 materials-16-04045-f016:**
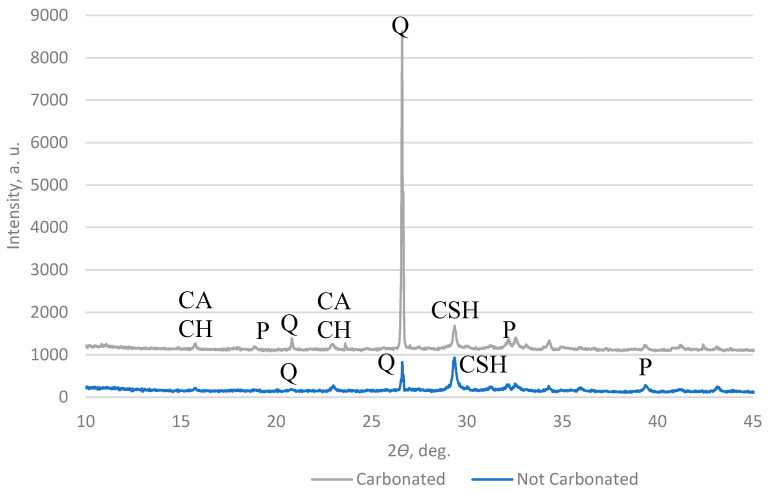
XRD pattern of AA2 (Q—quartz; P—portlandite; CACH—calcium aluminate carbonate hydrate; CSH—calcium aluminate hydrate).

**Figure 17 materials-16-04045-f017:**
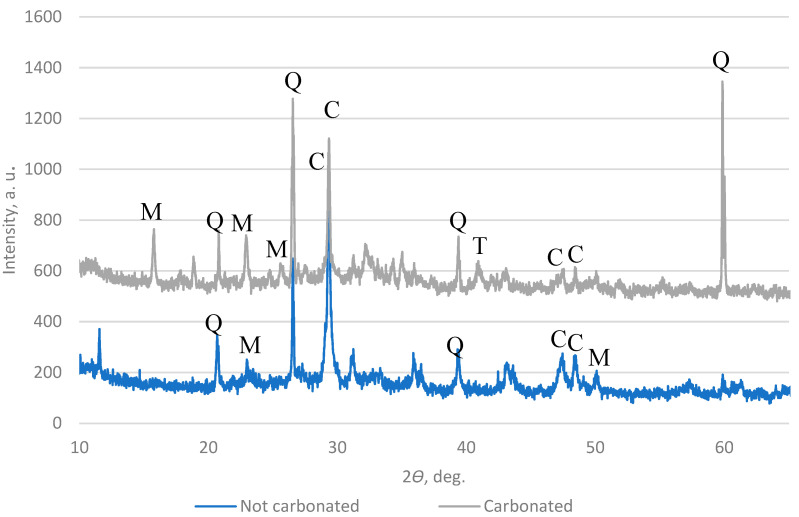
XRD pattern of AA3 (Q—quartz; C—calcite; T—calcium magnesium aluminum oxide silicate; M—alite).

**Figure 18 materials-16-04045-f018:**
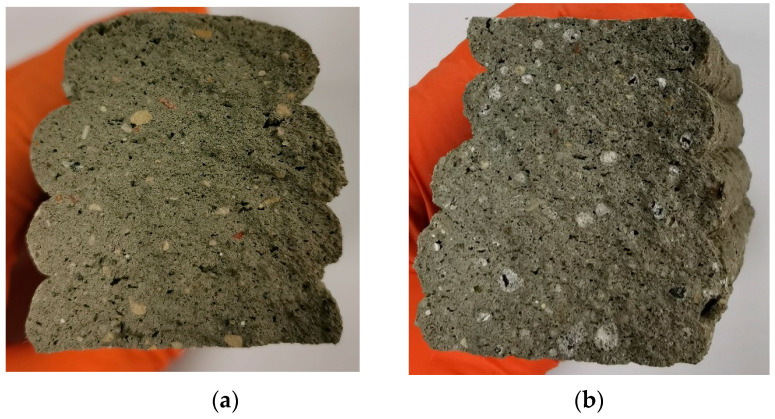
Printed concrete element cross-sections for printed composite numbers (**a**) LV1, (**b**) LV2, (**c**) LV3, and (**d**) LV4.

**Figure 19 materials-16-04045-f019:**
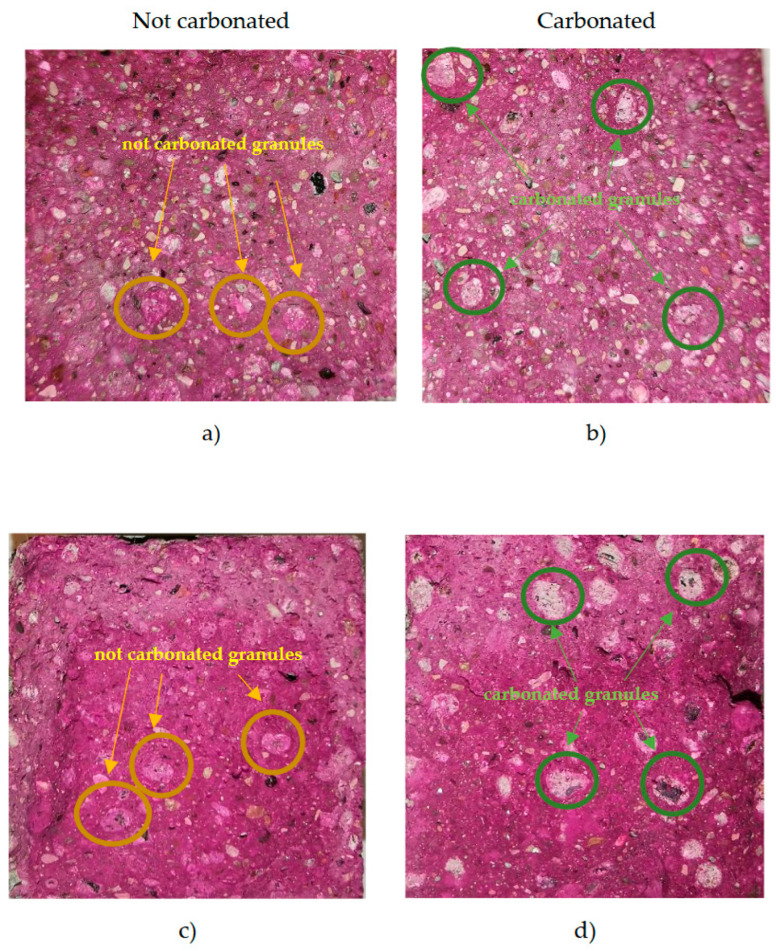
Results of the phenolphthalein spray test on 3D printed concrete LV2 with uncarbonated AA (**a**), LV2 with carbonated AA (**b**), LV3 with uncarbonated AA (**c**), and LV3 with carbonated AA (**d**).

**Figure 20 materials-16-04045-f020:**
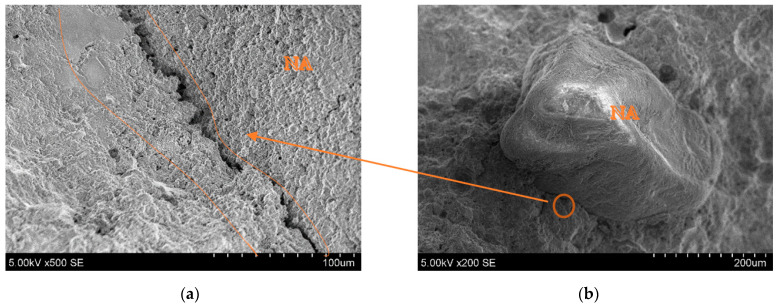
Microscopic (SEM) images of printed objects: (**a**) cement matrix connection zone between NA and printed object; (**b**) NA position in crosscut.

**Figure 21 materials-16-04045-f021:**
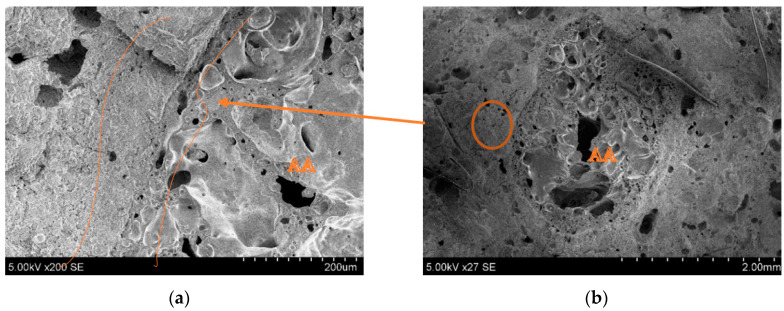
Printed objects containing AA. SEM images (**a**,**c**,**e**) are connection zones between AA and the cement matrix; (**b**,**d**,**f**) are AA in printed element crosscuts.

**Table 1 materials-16-04045-t001:** Chemical compositions and density of used binders.

Material	SiO_2_ (%)	Al_2_O_3_ (%)	Fe_2_O_3_ (%)	CaO (%)	MgO (%)	SO_3_ (%)	Na_2_O (%)	K_2_O (%)	TiO_2_ (%)	Mn_2_O_3_ (%)	P_2_O_5_ (%)	Cl (%)	CO_2_ (%)	Density kg/m^3^
Bottom slag (BS)	46.11	7.66	9.46	20.14	2.65	3.71	3.01	1.27	1.17	0.14	2.11	0.89	-	2500–2600
Burnt fly ash (BFA)	27	7	4–5	45–51	4–5	9	0.15	3–4	-	-	-	0.47	-	2700–2900
Portland cement CEM I 42.5 R (OPC)	18–20	4–5	3–4	62–65	3–4	3.3	0.1	1–1.5	-	-	-	-	-	2750–3200
Hydrated lime (HL)	-	-	-	94–96	0.3–0.4	0.05–0.10	-	-	-	-	-	-	0.5–4.0	2240

**Table 2 materials-16-04045-t002:** Artificial aggregate composition design.

Materials	Artificial Aggregate Name and Amount of Materials (kg/m^3^)		
	AA1	AA2	AA3
Bottom slag waste	1211.4	1148.5	1201.0
Portland cement CEM I 42,5 R	484.6	-	12.1
Calcium hydroxide	-	459.4	-
Burnt shale ash	-	-	360.3
Water	363.4	344.5	360.3

**Table 3 materials-16-04045-t003:** Particle size determination of sieved AA.

AA Name	Sieve Mesh Diameter, mm and Residue in %						
	4.0	2.8	2.0	1.0	0.5	0.25	0.0
AA1	2.0	43.6	21.5	23.0	5.4	1.3	2.6
AA2	1.6	32.1	31.1	29.6	4.4	0.6	0.6
AA3	3.5	46.0	23.5	18.9	4.8	1.7	1.6
Sand 0–2	0.0	0.0	3.2	37.7	33.8	3.6	1.7

**Table 4 materials-16-04045-t004:** Sieved granules, average bulk densities.

Bulk Density, kg/m^3^	Artificial Aggregate Name		
	AA1	AA2	AA3
Noncarbonated granules	1100	1025	1020
Carbonated granules	950	1035	1035

**Table 5 materials-16-04045-t005:** AA measured particle density.

Density kg/m^3^	Artificial Aggregate Name		
	AA1	AA2	AA3
Not carbonated	1550	1460	1460
Carbonated	1500	1670	1610

**Table 6 materials-16-04045-t006:** AA crushing strength test results.

	Artificial Aggregate Name		
F_max_—kN	**AA1**	**AA2**	**AA3**
Not carbonated	7.95	10.05	4.95
Carbonated	13.9	18.4	12.8
Escalation, %	+74.8	+83.1	+158.6

**Table 7 materials-16-04045-t007:** EDS analysis of split granule and granule surface.

Element	2, Uncarbonated	2, Carbonated	2, Uncarbonated, Split	2, Carbonated, Split	3, Uncarbonated	3, Carbonated	3, Uncarbonated, Split	3, Carbonated, Split
	At%							
C	3.42	7.42	2.35	5.97	3.43	9.81	7.69	2.33
O	64.25	65.21	60.4	61.93	48.12	60.97	59.40	57.50
Fe	0.57	0.49	0.6	0.72	1.02	0.65	0.84	0.46
Ti	0.18	-	0.51	0.63	0.43	-	0.59	-
Ca	22.28	19.73	21.89	15.23	26.66	20.71	9.78	8.23
K	0.07	0.10	0.31	0.84	1.80	0.77	1.80	4.04
Cl	0.21	0.06	0.41	0.61	0.77	0.41	0.52	0.23
S	0.74	0.91	2.4	1.13	3.45	1.02	0.22	-
P	-	-	0.11	0.35	0.17	-	0.46	-
Si	5.89	4.47	6.52	5.43	6.56	3.10	10.93	20.23
Al	1.71	1.04	1.94	3.49	6.33	1.46	3.23	3.28
Mg	0.29	0.22	0.65	2.19	0.96	0.77	1.59	0.69
Na	0.40	0.30	0.65	1.5	0.29	0.34	2.94	3.01
Ba	-	-	1.28	-	-	-	-	-
Total	100.00	100.00	100.00	100.00	100.00	100.00	100.00	100.00

**Table 8 materials-16-04045-t008:** Granule strength results in etalon composite.

Composite Name	Carbonated Granules	Flexural Strength, MPa	Compressive Strength, MPa
LV2	Not	5.30	40.80
LV2	Yes	5.90	44.40
LV3	Not	6.20	43.30
LV3	Yes	5.80	44.40

**Table 9 materials-16-04045-t009:** Three-dimensional printing composite design.

Materials	Composite Name and Materials Amount in %			
	LV1	LV2	LV3	LV4
Portland cement CEM I 42,5 R	30.0	30.0	30.0	30.0
Sand 0–2 mm	55.0	41.0	41.0	27.5
Calcium hydroxide	2.0	2.0	2.0	2.0
Burnt shale ash	9.0	9.0	9.0	9.0
Additives	4.0	4.0	4.0	4.0
Carbonated granules AA2	-	14.0	-	-
Carbonated granules AA3	-	-	14.0	27.5

**Table 10 materials-16-04045-t010:** Fresh 3D printing composite parameters.

Composite Number	Water Amount, L/kg	Consistency/Flow, cm	Bulk Density, kg/m3	Air Content, %
LV1	0.145	16.4	2100	7.2
LV2	0.150	16.1	2050	7.6
LV3	0.154	17.5	2090	6.8
LV4	0.164	17.3	2010	6.8

**Table 11 materials-16-04045-t011:** Strength measurement results.

Composite Number	Printed and Cut Elements				Hand-Made Prisms 40*40*160 mm		
	Perpendicularly		Longitudinal		Flexural Strength MPa	Compressive Strength MPa	Density after 28 Days
	Flexural Strength MPa	Compressive Strength MPa	Flexural Strength MPa	Compressive Strength MPa			
LV1	5.2	14.5	6.6	6.9	7.9	56.9	2080
LV2	4.3	13.1	5.0	5.3	6.8	56.3	2050
LV3	5.0	14.0	6.9	7.5	7.4	49.3	1950
LV4	2.7	5.7	3.1	4.5	6.3	40.5	1940

## Data Availability

Not applicable.

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
