# Peer review of "Influence of Carbonated Bottom Slag Granules in 3D Concrete Printing"

_materials, 2023, doi:10.3390/ma16114045_

Round 1

Reviewer 1 Report

The idea of this paper is good, using artificial aggregates to replace natural aggregates for low carbon applications of 3D printed concrete in terms of raw materials. However, unfortunately, the quality of this study is not satisfactory. The authors' research idea is not clear, and the majority of chapters are spent on introducing the properties of artificial aggregates, while the effect of incorporating artificial aggregates on the 3D printed concrete properties is not investigated. Overall, it is like a technical report rather than a scientific paper.

1)    The introduction is almost entirely describing artificial aggregates, with very little introduction related to 3D printed concrete. Most importantly, the significance of the artificial aggregate application in 3D printed concrete is not explained, and I think the combined application of the two is very far-fetched.

2)    Other important indicators such as the water absorption of artificial aggregates in the experimental materials are not measured

3)   The article's structure is confusing and I don't think it is a scientific paper written logically, probably more like a lab report.

4)   The topic of this article is xx, but almost all of it is about the properties of artificial aggregates. the 3D printed concrete part is actually only about the mechanical properties, and frankly, I have some doubts whether the mechanical properties tests were actually done because there are no real pictures about the mechanical tests and there is very little description of this.

Author Response

Thank you for yours comments it helps me to understand where are  mistakes.

Reviewer 2 Report

There are some weaknesses through the manuscript which need improvement. Therefore, the current study cannot be accepted for publication in this form, but it has a chance of acceptance after a major revision. My comments and suggestions are as follows:

1- Abstract gives information on the main feature of the performed study, but some details about the conducted tests in this study must be added.

2- Authors must clarify necessity of the performed research. Objectives of the study must be clearly mentioned in introduction.

3- The literature study must be enriched. In this respect, authors must read and refer to the following papers: (a) https://doi.org/10.3390/su14159782 (b) https://doi.org/10.1016/j.conbuildmat.2022.128559 and other relevant research works.

4- Novelty needs to be explicit in highlights and abstract.

5- Broaden and update the literature review to better connect to the current effort in the field in the context of concrete 3D printing.

6- All figures must be illustrated in a high quality. In addition, figures require more fidelity and density of information.

7- Authors must add values of printing parameters to the revised version.

8- In its language layer, the manuscript should be considered for English language editing. There are sentences which have to be rewritten.

9- The conclusion must be more than just a summary of the manuscript. List of references must be updated based on the proposed papers. Please provide all changes by red color in the revised version.

Author Response

Thank you for your comments, revision and all your comments were added to manuscript.

Reviewer 3 Report

Dear Authors,

Thank you for your manuscript. If the research concept, material design & execution with a number of results makes this paper interesting to read, broken language description makes it hard to understand. 

In general, you need to improve your English writing skills and rewrite a good part of your paper, it is difficult to read it and understand what you want to say. Abstract must be elaborated as follows: problem, solution&novelty, conclusions. Introduction is too general, some of sentences must be rephrased&combined to have an easy reading. Please make it shorter, combine sentences and avoid general description. Please clearly state objectives and problems of the study, novelty of your paper. 

Lines 131-132: "Including fact that slag consists of from mostly unreactive material by itself in all granule composites were added binder or binders mentioned before" -> it is not clear what you want to say here? 

Lines 222-224: please add exact reference

Line 242: the same laboratory conditions? Which ones?

Please skip phrasing "including this fact"

Figure 13: not clear what you indicate by 1, 2 and 3. There are only 3 granules?! WA is rather high, can you explain why?

Figure 14: granules name? 

Where are you stdev in graphs? Can you please format your charts in the style of scientific publication and not in style of master thesis?

Lines 401-402: " In comparison with carbonated and not carbonated AA were taken pictures of granule and spited granule, to analyse inside of formatted aggregate also." What do you mean here?

SEM analysis requires a thorough analysis description.

Line 426: how humidity was measured?

Lines 425-439: the style of this paragraph differs a lot from your whole paper; well-written should be followed as example.

Overall, results description are quite hectic, too short. A more structural explicit approach should applied in description and switch between one group of results to another. 

Conclusions must be elaborated.

Author Response

Thank you for your comments, it will help me improve my scientific paper writing. I mark changed (rewritten) places in manuscript in red.

Round 2

Reviewer 2 Report

The paper has been improved and corresponding modifications have been conducted. In my opinion, the current version can be considered for publication.

Author Response

Thank you for comment, few more changes were done to improve level of manuscript.

Reviewer 3 Report

Please bring clear structure in your manuscript, make it better readable and with logic sequence on the results presentation.

Please elaborate sentences in lines 10-15. Abstract should have clear structure: problem, novelty and solution.

Line 46: bottom slag (ash)?! In general, slag has a common name as GGBFS. Slag is a by-product of smelting (pyrometallurgical) ores and used metals. Broadly, it can be classified as ferrous (by-products of processing iron and steel), ferroalloy (by-product of ferroalloy production) or non-ferrous/base metals (by-products of recovering non-ferrous materials like copper, nickel, zinc and phosphorus).Ground granulated blast-furnace slag (GGBS or GGBFS) is obtained by quenching molten iron slag (a by-product of iron and steel-making) from a blast furnace in water or steam, to produce a glassy, granular product that is then dried and ground into a fine powder.Ground granulated blast furnace slag is a latent hydraulic binder forming calcium silicate hydrates (C-S-H) after contact with water.

And what you have in your study is MSW bottom ash. There is difference between two terms and you use them together !? Please update your terminology in title and whole manuscript.

Your Introduction is still very general & with no structure. You can remove half of it and write it in a more saturated way adding certain references to already published studies and bring some good comparison to it. There should be logic line abstract-intro-results discussion- conclusions.

Figures 11 & 12, please adjust stdev, those cannot be the same (investigate please excel options for it).

Whenever you use abbreviations, please make sure that definition is provided earlier, for example, in line 297 it appears for the first time.

Figures 19 (a, b, d) are out of focus photos.

Figure 21, at first there should be indication of a, b, c etc in your figure caption. Secondly, it is advisable to give your sem description to read to specialist who performs imaging before resubmission next time. Maybe that type of description is ok for master thesis but definitely not for research paper.

Author Response

Thank you for your comments, manuscript and English language was revised, structure changed.

Round 3

Reviewer 3 Report

bigger team - better quality, well done!

Introduction is still the weakest part of your paper, it has general wording and volume, half can be removed. In lines 534-540 provided interesting outcome of your paper, focus on it in your Intro. Please don't use "we", "some investigations", etc. At the last paragraph clearly indicate the goal of your paper and what you bring along with it in comparison to what is already published or not published... 

make sure you don't have ashes and slags still mixed in your description.

line 288: should be -> 'developed within Riga Technical University (RTU)'

lines 493-523: please elaborate description, another weak part of your paper

Conclusions: first paragraph the goal of the study & novelty followed by achieved outcome in a bullet form.

Please format your list of references according to the journal requirements

Author Response

Thank you for comments, it has been addressed and edited.
